# Identity-by-descent with uncertainty characterises connectivity of *Plasmodium falciparum* populations on the Colombian-Pacific coast

**Aimee R. Taylor** [1,2]*, **Diego F. Echeverry** [3,4,5], **Timothy J. C. Anderson** [6], **Daniel E. Neafsey** [2,7], **Caroline O. Buckee** [1]

**1** Center for Communicable Disease Dynamics, Department of Epidemiology, Harvard T. H. Chan School of Public Health, Boston, Massachusetts, USA, **2** Infectious Disease and Microbiome Program, Broad Institute of MIT and Harvard, Cambridge, Massachusetts, USA, **3** Centro Internacional de Entrenamiento e Investigaciones Médicas (CIDEIM), Cali, Colombia, **4** Universidad Icesi, Calle 18 No. 122-135, Cali, Colombia, **5** Departamento de Microbiologia, Facultad de Salud, Universidad del Valle, Cali, Colombia, **6** Disease Intervention and Prevention Program, Texas Biomedical Research Institute, San Antonio, Texas, USA, **7** Department of Immunology and Infectious Diseases, Harvard T. H. Chan School of Public Health, Boston, Massachusetts, USA

* ataylor@hsph.harvard.edu

**Data Availability Statement:** All data analyses were performed in R. The data are publicly available as a .RData files and the code is publicly

## Abstract

Characterising connectivity between geographically separated biological populations is a common goal in many fields. Recent approaches to understanding connectivity between malaria parasite populations, with implications for disease control efforts, have used estimates of relatedness based on identity-by-descent (IBD). However, uncertainty around estimated relatedness has not been accounted for. IBD-based relatedness estimates with uncertainty were computed for pairs of monoclonal *Plasmodium falciparum* samples collected from five cities on the Colombian-Pacific coast where long-term clonal propagation of *P. falciparum* is frequent. The cities include two official ports, Buenaventura and Tumaco, that are separated geographically but connected by frequent marine traffic. Fractions of highly-related sample pairs (whose classification using a threshold accounts for uncertainty) were greater within cities versus between. However, based on both highly-related fractions and on a threshold-free approach (Wasserstein distances between parasite populations) connectivity between Buenaventura and Tumaco was disproportionally high. Buenaventura-Tumaco connectivity was consistent with transmission events involving parasites from five clonal components (groups of statistically indistinguishable parasites identified under a graph theoretic framework). To conclude, *P. falciparum* population connectivity on the Colombian-Pacific coast abides by accessibility not isolation-by-distance, potentially implicating marine traffic in malaria transmission with opportunities for targeted intervention. Further investigations are required to test this hypothesis. For the first time in malaria epidemiology (and to our knowledge in ecological and epidemiological studies more generally), we account for uncertainty around estimated relatedness (an important consideration for studies that plan to use genotype versus whole genome sequence data to estimate IBD-based relatedness); we also use threshold-free methods to compare parasite populations

available as .R scripts at https://github.com/artaylor85/ColombianBarcode.

**Funding:** A.R.T. and C.O.B. are supported by a Maximizing Investigators' Research Award for Early Stage Investigators (R35 GM-124715) (https://www.nih.gov/). D.F.E. received financial support from Colciencias, call 656-2014 "Es Tiempo de Volver" award FP44842-503-2014 (https://minciencias.gov.co/). T.J.C.A. is supported by funds from the National Institute of Allergy and Infectious Diseases, National Institutes (R37 AI048071) (https://www.niaid.nih.gov/). This project was funded in part with federal funds from the National Institute of Allergy and Infectious Diseases, National Institutes of Health, Department of Health and Human Services, under grant number U19 AI-110818 to the Broad Institute (D.E. N.) (https://www.niaid.nih.gov/). The funders had no role in study design, data collection and analysis, decision to publish, or preparation of the manuscript.

**Competing interests:** The authors have declared that no competing interests exist.

and identify clonal components. Threshold-free methods are especially important in analyses of malaria parasites and other recombining organisms with mixed mating systems where thresholds do not have clear interpretation (e.g. due to clonal propagation) and thus undermine the cross-comparison of studies.

## Author summary

In this study we aimed to characterise connectivity between populations of *Plasmodium falciparum* malaria parasites sampled from five cities on the Colombian-Pacific coast where long-term clonal propagation of *P. falciparum* is frequent. We found that connectivity along the coast is consistent with accessibility not isolation-by-distance, potentially implicating marine traffic in malaria transmission and thus presenting a possible opportunity for targeted intervention. Our study makes methodological contributions that could be adapted to analyses of other recombining organisms. Akin to numerous studies in both epidemiology and ecological, to characterise connectivity, we used genetic data and computed estimates of relatedness based on identity-by-descent (IBD). However, unlike previous studies, confidence intervals around relatedness estimates were included in our analyses. This is an important consideration for all studies that plan to use limited genetic data to estimate IBD-based relatedness. To identify groups of clonal parasites and to compare parasite populations across cities, we used methods that avoid thresholds, e.g. of highly-related parasite pairs. Threshold-free methods promote cross-comparison in studies of recombining organisms for which thresholds do not have a clear interpretation (e.g. for malaria parasites, where the frequency of clonal propagation varies in space and time and is not fully understood).

## Introduction

In many research fields genetic data are used to help characterise connectivity between geographically distinct biological populations, with numerous applications in conservation, agriculture, and public health. Patterns of genetic similarity between pathogen populations help us understand how the disease spreads. Patterns of relatedness (a measure of genetic similarity) between malaria parasites sampled from different human populations, for instance, help characterise the connectivity between different malaria parasite populations, thus guide the design of targeted public health interventions [1].

Several methods are employed to measure genetic similarity and thus characterise connectivity. Phylogenetic methods, in which genetic distances between individuals are measured in units of mutation [2], are most applicable to rapidly mutating organisms that do not recombine (e.g RNA viruses) [3]. Studies of relatedness, in which relatedness is a measure of probability of inter-individual identity-by-descent (IBD), are applicable to organisms that do recombine (e.g. malaria parasites). Population genetic parameters of allelic variation (e.g. $F_{ST}$) are applicable to all organisms (those that do and do not recombine), but do not generate measures of genetic distance or similarity on an inter-individual level, thus provide less granularity. Moreover, among recombining organisms, inter-population allelic variation tends to accumulate more slowly than inter-individual variation in IBD [4]. As such, analyses of relatedness sometimes recover evidence of nearby and recent connectivity where analyses of $F_{ST}$ do not [5].

Malaria parasites are protozoan parasites that undergo an obligate stage of sexual recombination in the mosquito midgut. Like many organisms (e.g. many plants [6, 7]), malaria parasites have a mixed mating system that encompasses both inbreeding and outcrossing. The extent to which malaria parasites outcross depends on transmission intensity and is not fully understood [8]. For outcrossing to occur a mosquito must ingest genetically distinct gametocytes. Humans can be infected by multiple genetically distinct parasite clones that are either co-transmitted via inoculation from a single mosquito, in which case they are likely recombinants so inter-related (unless they derive from different blood meals), or transmitted independently by multiple mosquitoes (a mechanisms coined superinfection by George MacDonald, 1950 [9, 10]), in which case the parasite clones are likely unrelated [11, 12]. The latter can occur in a setting where the entomological inoculation rate is high; recent work suggests co-transmission is important in both low and high transmission settings [12].

Malaria genomic epidemiology studies of connectivity are increasingly common, especially in the context of public health and using genotype (versus whole genome sequence) data [5, 13–16]. Using IBD-based relatedness but not $F_{ST}$, evidence of isolation-by-distance among *P. falciparum* populations along a 100 km stretch of the Thailand-Myanmar border was found [5]. This study was based, in part, on analyses of monoclonal *P. falciparum* samples genotyped at 93 single nucleotide polymorphisms (SNPs). Based on $F_{ST}$ estimated using *P. falciparum* samples genotyped at 250 SNPs, a different study found evidence of departure from isolation-by-distance among *P. falciparum* populations along a 500 km stretch of the Colombian-Pacific coast where transmission is mixed (low but high in some regions) and outcrossing limited [13, 17]. In the current study, we re-explore this departure from isolation-by-distance with more granularity using IBD-based relatedness. For the first time in malaria epidemiology (and, to our knowledge, for the first time in ecological and epidemiological studies more generally), we account for uncertainty in relatedness estimates; we also use threshold-free methods to compare parasite populations and identify clonal components. The original study [13] is described in more detail below.

Malaria epidemiology in Colombia is associated with a multitude of ecological, evolutionary and social factors, including human migration due to deforestation, illegal crops, gold mining [18–22], and the mass emigration of people fleeing the humanitarian crisis in Venezuela [23–26]. Understanding the interplay between e.g. human migration, parasite population connectivity and the spread of antimalarial resistance is critical [18, 20]. For example, if resistance is driven by spread (versus de-novo mutation), targeted efforts to eliminate hotspots of transmission (e.g. in eastern Myanmar [27, 28]) may help to prolong the longevity of compromised antimalarial therapies. To ensure adequate isolation, thereby prevent re-population, units of targeted intervention need to account for parasite population connectivity, which relates to human migration [29–31]. In preparation for studies of resistance, Echeverry et al. genotyped *P. falciparum* samples from four provinces on the Colombian-Pacific coast [13]. Clonality, population structure and linkage disequilibrium (LD) were characterised using a suite of population genetic analyses. The results were highly informative: the vast majority of successfully genotyped *P. falciparum* samples were deemed monoclonal (325 of 400) with a strong association between incidence and clonality. Among the 325 monoclonal samples, 136 unique haploid multilocus genotypes (MLGs) were identified using relatedness based on identity-by-state (IBS), which is a correlate of IBD [32] (and has been used elsewhere to characterise connectivity between nearby malaria parasite populations [14–16]). Of the 136 MLGs, 44 infected two or more patients (max. 28 patients), 45 persisted for two or more days (max. 8 years), and 7 of the 15 most common MLGs were sampled in two or more provinces (max. all four provinces). Panmixia was rejected based on evidence of four sympatric but geographically structured subpopulations; and, overall, LD decayed at a rate that was faster than expected for South American *P. falciparum* populations (compare with e.g. [33]). Echeverry et al. concluded that

evidence of low genetic diversity, persistent MLGs and population structure is consistent with low transmission and limited outcrossing, while evidence of a relatively fast rate of LD decay and of shared MLGs across provinces is consistent with extensive human movement connecting *P. falciparum* populations.

Although the study by Echeverry et al. features analyses of IBS-based relatedness (i.e. MLGs), evidence of departure from isolation-by-distance was based on $F_{ST}$ alone. To explore isolation-by-distance in more granularity while accounting for uncertainty, we compute IBD-based relatedness estimates and confidence intervals for all pairs of 325 monoclonal parasite samples. Akin to previous studies (e.g. [5]), highly-related parasites were classified using a threshold; however, confidence intervals allow uncertainty to be accounted for in this study. For example, in [5] a parasite pair was considered highly-related if its relatedness estimate exceeded 0.5, whereas here a parasite pair is considered highly-related if the lower end-point of the 95% confidence interval around its relatedness estimate exceeds some stated value, which is 0.25 in the main text and 0.5 in sensitivity analyses. This is important because uncertainty can overwhelm relatedness estimated using limited genotype data [32]. Our approach includes two additional contributions. First, we complement our analysis of highly-related parasites with a threshold-free approach that uses a metric called the 1-Wasserstein distance, which can be interpreted as the cost of transporting a distribution of parasite samples from one city to another [18, 34]. Second, we identify groups of statistically indistinguishable parasites, which we call clonal components, using the simple concept of components from graph theory and confidence intervals. Confidence intervals circumvent reliance on an arbitrary clonal threshold (i.e. some number of differences tolerated between parasites samples considered clonal). Graph components circumvent reliance on unsupervised clustering methods that are sensitive to both the definition of genetic similarity and algorithmic specification [35, 36]. Overall, our approach could be adapted to viruses and bacteria that show recombination or reshuffling of segments as well as clonal propagation [37–40], to other protozoans (e.g. *Toxoplasma*, *Cryprosporidium* [41–43]), and to the many fungal pathogens [44], plants [6, 7], and animals with mixed mating systems. Due to our treatment of uncertainty, it is especially relevant for a growing number of studies that plan to estimate IBD-based relatedness using genotype (versus sequence) data.

## Results

### Relatedness estimates between *P. falciparum* sample pairs

For all 52650 pairwise comparisons of 325 previously published monoclonal *P. falciparum* samples with data on 250 biallelic single nucleotide polymorphisms (SNPs) [13], relatedness was estimated using the hidden Markov model (HMM) described in [32]. Relatedness is thus defined as the probability that, at any SNP, the two alleles drawn from the paired monoclonal *P. falciparum* samples are IBD.

The parasite samples were collected between 1993 and 2007 from symptomatic patients participating in studies at five cities on the Colombian-Pacific coast (S1 Table). Despite considerable uncertainty, all estimates are informative (Fig 1). That is to say, there are no relatedness estimates whose 95% confidence intervals span entirely from zero to one. The vast majority of relatedness estimates were classified unrelated.

### Highly-related *P. falciparum* sample pair fractions partitioned in space and time

In our main analysis (Fig 2), highly-related parasite samples were classified using an arbitrary threshold of 0.25 (Table 1), which corresponds to the expected relatedness between parasites

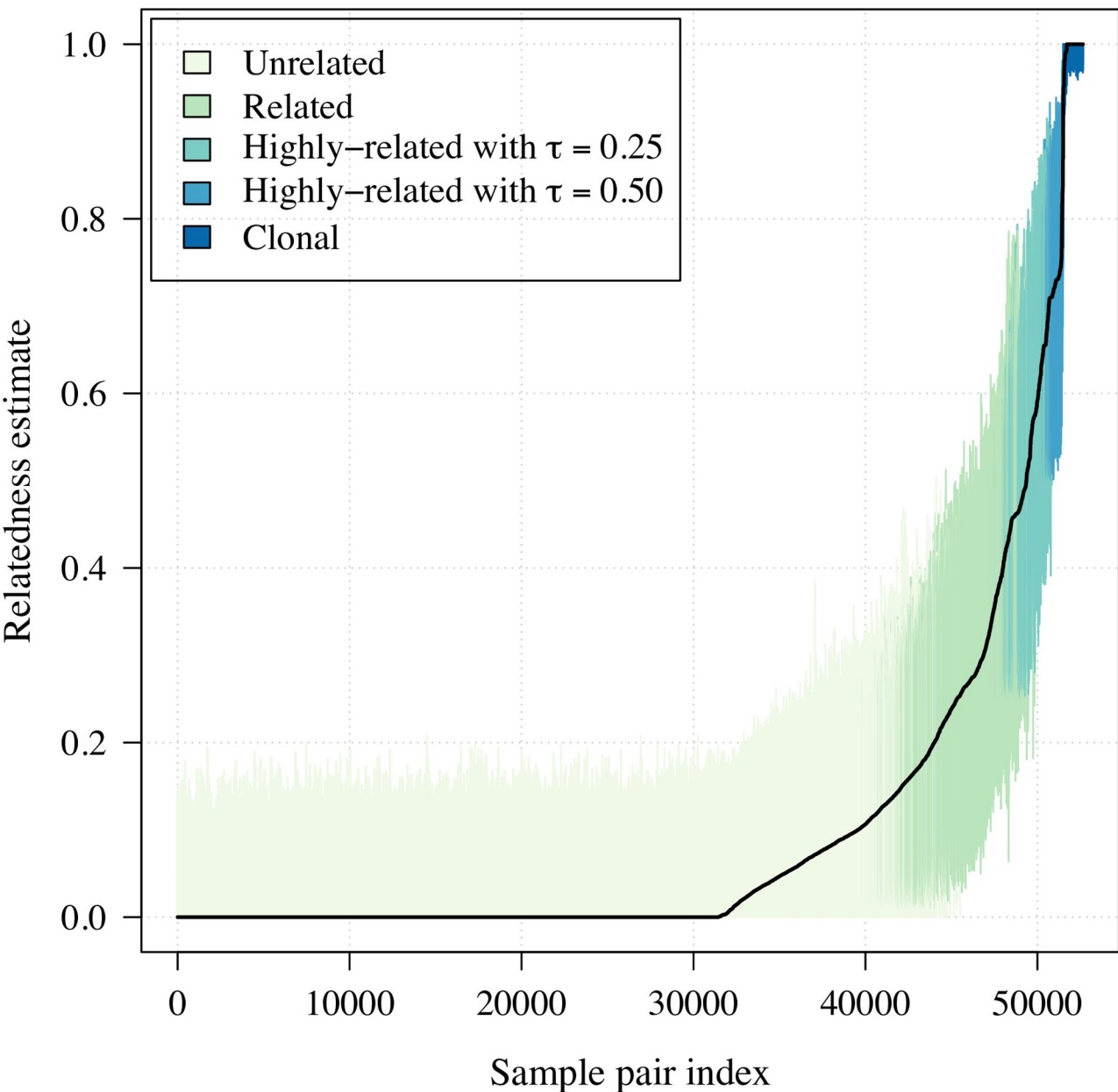

**Fig 1. Estimates of relatedness with 95% confidence intervals.** Estimates and confidence intervals are shown for all 325 choose two (52650) *P. falciparum* sample pairs and are ordered by increasing relatedness estimate. Confidence intervals are coloured according to classifications based on lower and upper confidence interval end-points, where $\tau$ is an arbitrary threshold used to classify highly-related pairs. For example, a pair is considered highly-related with $\tau = 0.25$ if the lower end-point of the confidence interval around its relatedness estimate exceeds 0.25. Otherwise stated, if its relatedness estimate is statistically distinguishable from 0.25.

separated by two outcrossed generations, but is hard to interpret in the context of frequent clonal propagation. Despite few highly-related *P. falciparum* sample pairs overall, there are three notable observations regarding their fraction partitioned in space and time. First, there is a greater fraction of highly-related sample pairs among those collected closer together in time (Fig 2(A)). Second, the fraction of highly-related sample pairs is generally greater within cities than between, with Guapi having the largest fraction of highly-related pairs and Buenaventura

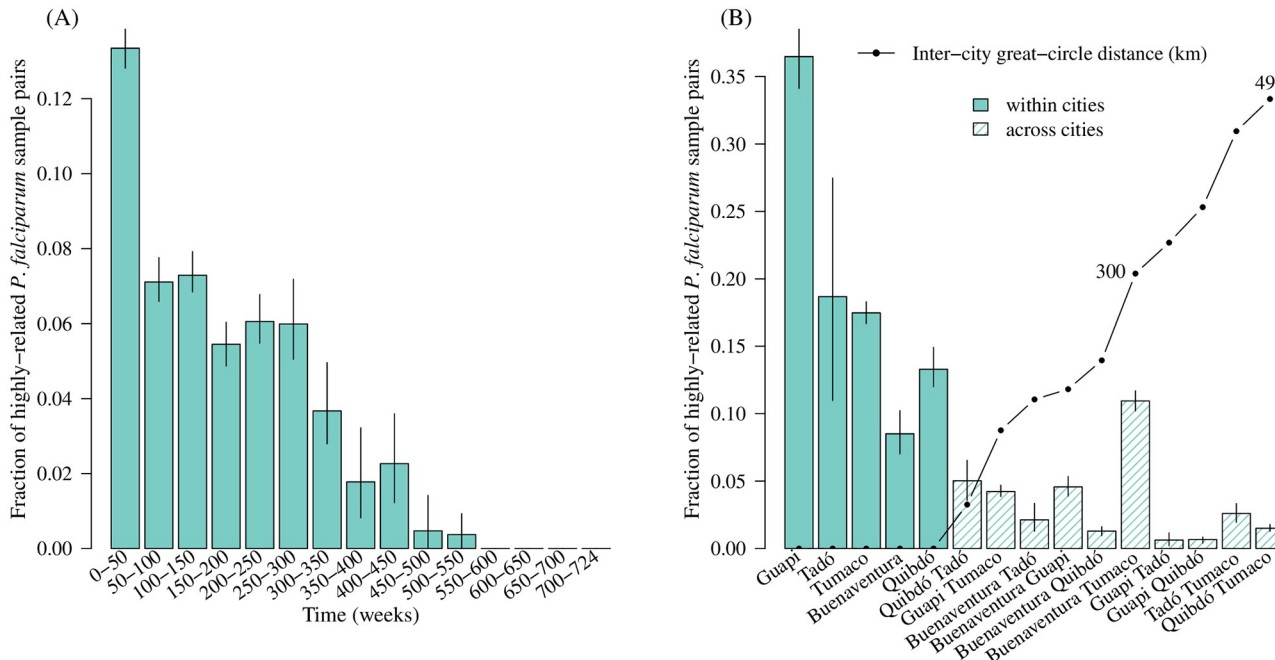

**Fig 2. Fractions of highly-related sample pairs partitioned in time and space.** (A) Partitioned by time between collection dates. (B) Partitioned by collection city, where the inter-city great-circle distance is the distance in kilometres (km) between city pairs on the Earth's surface.

having the lowest (Fig 2(B)). However, third, the fraction shared between Buenaventura and Tumaco is greater than expected given inter-city distance (Fig 2(B)). These observations are largely robust to different high-relatedness thresholds (S1 Fig). Spatial trends evaluated using a threshold-free approach are also consistent: they show a general increase in 1-Wasserstein distance with inter-city distance besides Buenaventura and Tumaco (Fig 3). The 1-Wasserstein distance can be interpreted as the total cost required to transport a distribution of parasite samples from one city to another [18, 34], where the cost of transporting a single parasite to another is equal to one minus relatedness. The small 1-Wasserstein distance between Buenaventura and Tumaco is thus consistent with elevated gene flow between *P. falciparum* populations sampled from these cities.

Fig 4 shows the inter-city *P. falciparum* population connectivity of Fig 2(B) projected onto a map of the Colombian-Pacific coast. Buenaventura and Tumaco are the two largest official ports on the Colombian-Pacific coast (Buenaventura is the largest) and are connected by frequent marine traffic (www.marinetraffic.com). Although Tumaco is connected to Buenaventura via the Pan-American highway, which connects all sites but Guapi, primary access to Tumaco is via the port due to difficult and unsafe country roads in Nariño.

**Table 1. Classification of parasite sample pairs.** Classification is based on the lower and upper end-points (LCI and UCI, respectively) of the 95% confidence interval around each relatedness estimate, $\hat{r}$, where $\epsilon$ is an arbitrarily small number to identify LCI ≈ 0 and UCI ≈ 1 given that LCI and UCI ∈ (0, 1) not [0, 1]; and $\tau$ is an arbitrary threshold used to classify highly-related pairs. We use $\epsilon = 0.01$ throughout, $\tau = 0.25$ (main analysis) and $\tau \in \{0.25, 0.50\}$ (sensitivity analysis).

| Classification | Interpretation | Definition |
|---|---|---|
| Unrelated | $\hat{r}$ statistically indistinguishable from zero | LCI < $\epsilon$ |
| Related | $\hat{r}$ statistically distinguishable from zero | LCI > $\epsilon$ |
| Highly-related | $\hat{r}$ statistically distinguishable from a specified threshold | LCI > $\tau$ |
| Clonal | $\hat{r}$ statistically indistinguishable from one | UCI > 1 − $\epsilon$ |

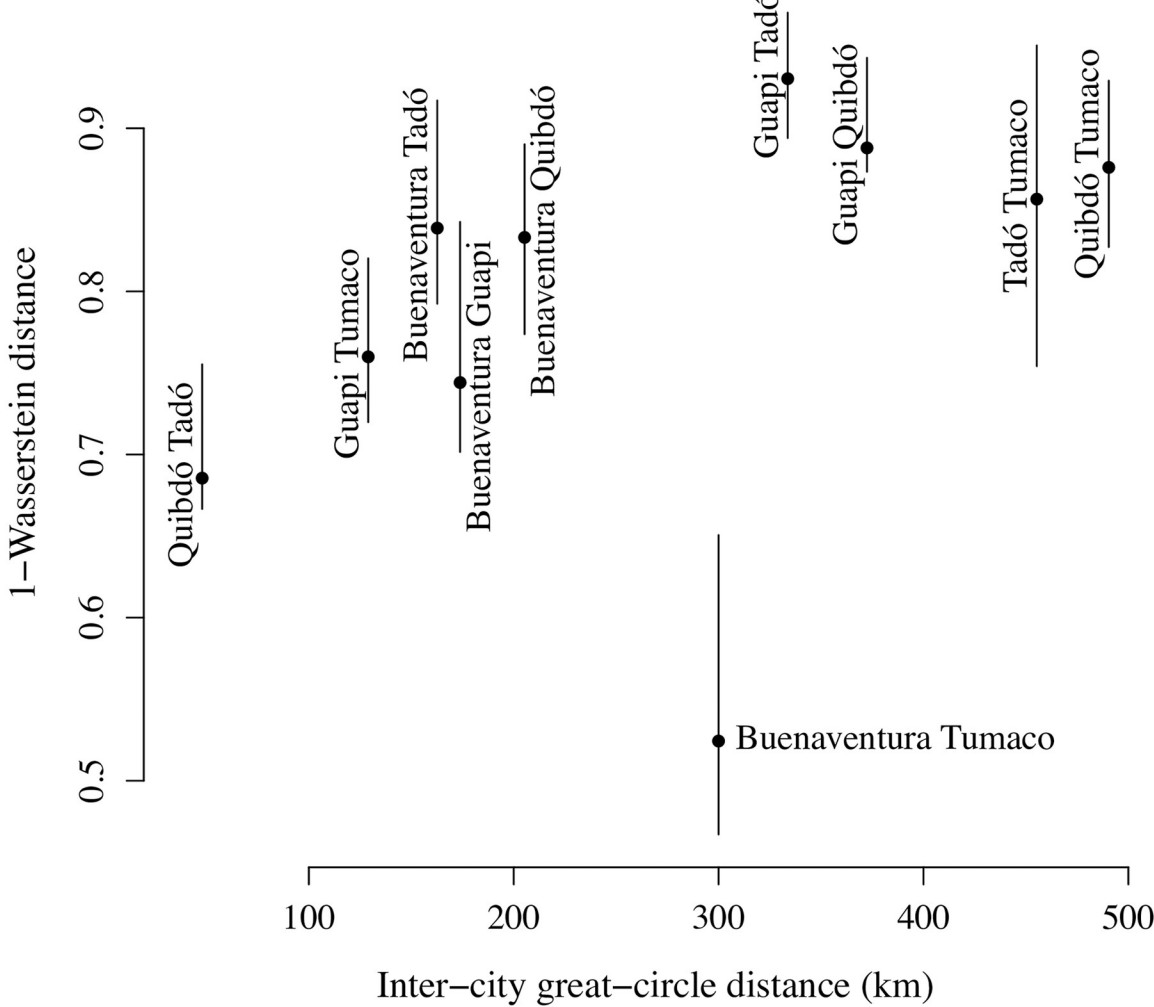

**Fig 3. *P. falciparum* population connectivity assessed using a threshold-free approach.** 1-Wasserstein distance between parasite populations from different cities versus inter-city great-circle distance in kilometres (km).

Guapi, which is effectively unreachable by road and not an official port, is connected by marine traffic but with less frequency (www.marinetraffic.com). Consistent with its isolation, the fraction of highly-related parasite pairs is relatively large within Guapi (Fig 2(B)), and very small between Guapi and the two inland cities, Quibdó and Tadó (Figs 2(B) and 4). Moreover and importantly regarding the elevated fraction of highly-related samples pairs within both Guapi and Tadó (Fig 2(B)), all samples from Guapi and Tadó were collected within a single year (S1 Table). The low fraction of highly-related parasite sample pairs within Buenaventura (Fig 2(B)) is in part consistent with it having contributed samples over many years (S1 Table) and with it being the most important port on the Pacific coast (www.marinetraffic.com), i.e. a hub through which human traffic and thus potential parasite mixing is high [13].

The apparent association between *P. falciparum* population connectivity and the frequency of marine traffic raises questions about the latter's role in malaria transmission. However, other scenarios could lead to these relationships, for example high connectivity could result from a single travel event between Buenaventura and Tumaco, followed by expansion of

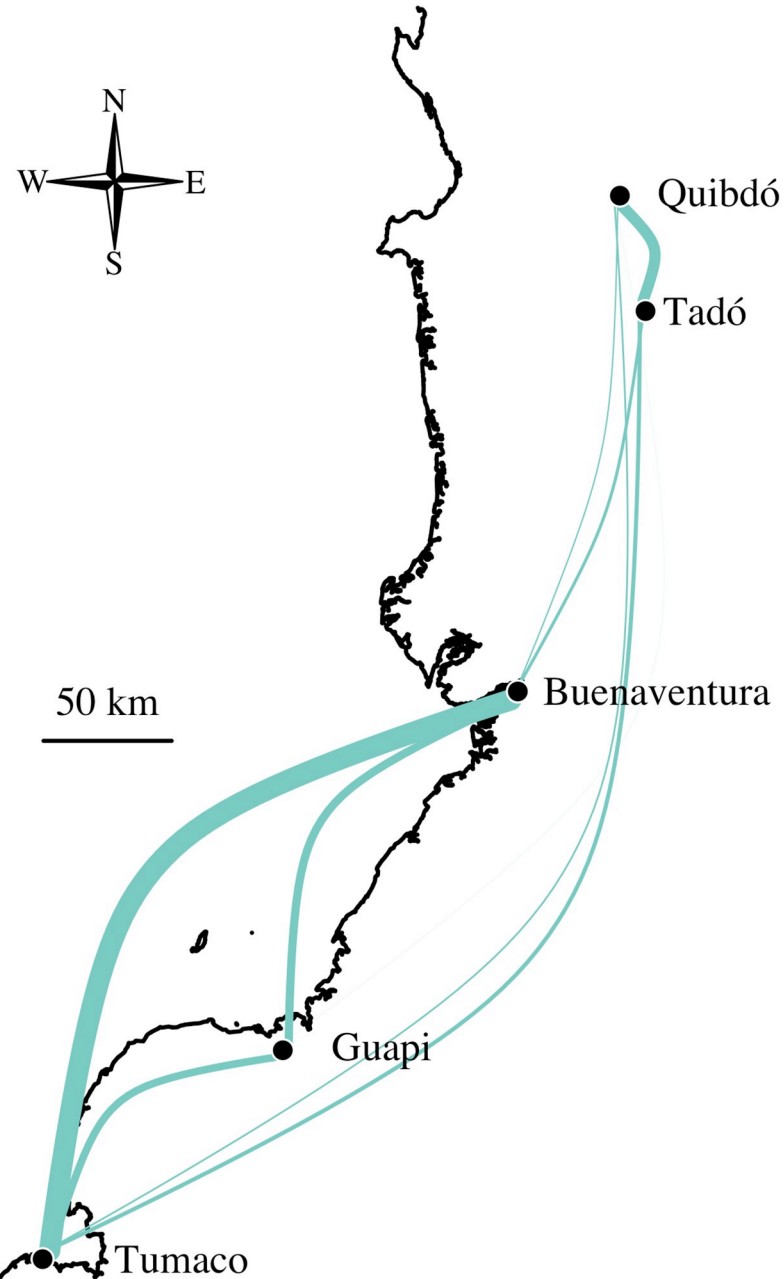

**Fig 4. *P. falciparum* population connectivity based on fractions of highly-related sample pairs.** The width of each inter-city edge is proportional to the fraction of highly-related sample pairs across cities plotted in Fig 2(B). Note that the edges between Guapi and Quibdó and Guapi and Tadó are plotted but too thin to visually discern.

highly-related and clonal parasites. To further explore the genetic signal that supports this association we next consider clonal components.

## Clonal components

We define clonal components as groups of statistically indistinguishable parasite samples identified under a graph theoretic framework: consider a graph whose vertices are parasite samples

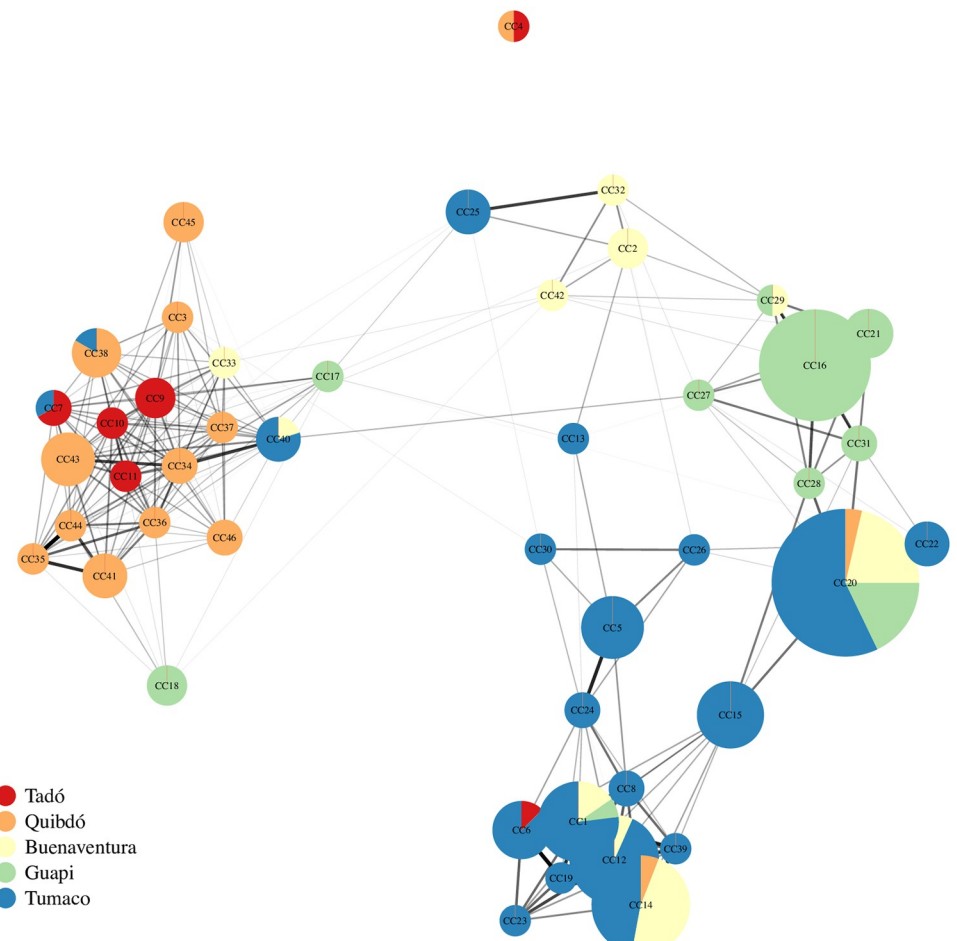

**Fig 5. Clonal components and the average relatedness between them.** Vertices depict clonal components, which are groups of two or more statistically indistinguishable parasite samples. CC vertices are plotted using the Fruchterman-Reingold layout algorithm [45], thereby clustering inter-related CCs. The size of each CC vertex is proportional to the number of parasite samples per CC, ranging from 2 to 28 statistically indistinguishable parasite samples. CCs are named in order of the collection date of the earliest parasite sample per CC (S2 Table). CCs with parasite samples collected from two or more cities are depicted as pie charts. Colour denotes the city of parasite sample collection. Edge transparency and weight is proportional to average relatedness, ranging from 0.003 to 0.840. Relatedness estimates that are indistinguishable from zero were set to zero. Edges whose average relatedness is zero are not plotted. Each CC besides CC4 is related to at least one other. CC4 contains two samples (one from Tadó, another from Quibdó). It is likely a contaminant; see main text.

and whose edges are weighted by relatedness estimates, a clonal component is a sub-graph whose vertices are all connected to one another via edges whose weights are statistically indistinguishable from one (i.e. clonally related, Table 1). In total, 46 distinct clonal components were detected, ranging in size from 2 to 28 statistically indistinguishable parasite samples (Fig 5). They are spatially clustered. Ten of the 46 contain parasite samples collected from two or more cities. Each clonal component besides one (clonal component four) is on average related to at least one other (Fig 5). The unrelated clonal component is almost certainly an artefactual contaminant: it accords with MLG 036 reported in [13], where contamination during *in vitro* adaptation or DNA manipulation was suspected (MLG 036 contained "two culture-adapted samples from Quibdó and Tadó that were indistinguishable from the Dd2 reference strain from Southeast Asia"—the Dd2 reference strain was included as a control when the data were originally generated [13]).

Clonal parasite samples detected in both Buenaventura and Tumaco belong to five distinct clonal components (1, 12, 14, 20 and 40, Fig 5). We thus dismiss a single travel event connecting Buenaventura and Tumaco involving a single parasite clone. We cannot dismiss a single travel event involving multiple parasite clones, however. Based on the proportions of multiclonal infections in the original data from Buenaventura and Tumaco (14% and 19% respectively) [13], the probability that these five clones could be distributed across infections in four or fewer individuals is approximately 0.6. Indeed, three of the five clonal components are interrelated on average (S3 Table). As such, they could derive from co-transmitted recombinant parasites transported in a single individual with a multiclonal infection. On the contrary, the remaining two clonal components have relatedness estimates that are not statistically distinguishable from zero. As such, they could derive from a single superinfected individual, or from different individuals with independent monoclonal infections. Unfortunately, the data required to further evaluate these scenarios (data on the multiplicity of multiclonal infections, and on relatedness within multiclonal infections, e.g. [12]) are not available. Given dates and cities of first detection (S2 Table), it is tempting to suggest some clonal components predate others and originate in specific locations. For example, it is possible that parasite samples from clonal components 1 and 20 in Buenaventura and Tumaco emanated from Guapi, creating a spurious link between Buenaventura and Tumaco. However, because these data are from sparsely sampled symptomatic cases in a setting where clonal propagation is frequent, sample collection chronology is not necessarily representative of the chronology of transmission chain events (S2 Fig).

Regarding transmission chain events, we note that clonal component 20 relates to the three inter-related clonal components (1, 12 and 14) via an intermediate clonal component detected in Tumaco only (clonal component 15) as well as an intermediate parasite sample from Quibdó that does not belong to a clonal component (S3 Fig). These intermediates likely derive from recombination between parasites related to the clonal components they connect. Several connections consistent with recombinants can be found among the relatedness graphs (Fig 5 and S3 Fig). As such, it seems it may be at least theoretically possible to construct approximate *P. falciparum* transmission chains given more dense sampling of malaria infections on the Colombian-Pacific coast.

## Discussion

Here we show that estimates of IBD-based relatedness and their associated uncertainty can be used to uncover evidence of epidemiologically meaningful connectivity between *P. falciparum* populations on a relatively local spatial scale: along the Colombian-Pacific coast where clonal propagation is frequent [13], extending southward to Ecuador [46, 47]. While our approach largely confirms a previous report based on $F_{ST}$ [13], estimates of relatedness provide more granularity while their confidence intervals account for uncertainty thus provide more statistical rigor, e.g. when highly-related parasite sample pairs are classified. Our approach includes two additional contributions: 1-Wasserstein distances are used to compare parasite populations in an entirely threshold-free manner; and clonal components are identified using graph components and confidence intervals, thereby circumventing reliance on an arbitrary clonal threshold. Threshold-free methods are especially important in analyses where thresholds do not have clear interpretations (e.g. 0.5 may correspond to the expected relatedness of siblings in an outcrossed population, but its interpretation is unclear in a population where inbreeding and clonal propagation is common) and thus undermine the cross-comparison of studies. Standardisation will accelerate the maturation of malaria genomic epidemiology and facilitate the translation of research into actionable insight for policy makers [1]. Our overall

approach could also be adapted for analyses of other recombining organisms with mixed mating systems.

IBD-based relatedness estimates recovered 1) a large fraction of highly-related parasite sample pairs within Guapi, a city on the Colombian-Pacific coast that is relatively isolated besides infrequent marine traffic; 2) a low fraction of highly-related parasite sample pairs within Buenaventura, the most important port on the Colombian-Pacific coast and thus the least isolated city in this study; and 3) a disproportionally large fraction of highly-related parasite pairs between Buenaventura and Tumaco (departure from isolation-by-distance), where Tumaco is the second largest port on the Colombian-Pacific coast. These observations accord with several published previously: 1) elevated LD in a *P. falciparum* subpopulation (identified using STRUCTURE [35, 48]) predominant in Guapi; 2) rapid LD decay in a *P. falciparum* subpopulation predominant in Buenaventura; and 3) lowest genetic differentiation (based on $F_{ST}$ estimates) between provinces Valle (Buenaventura) and Nariño (Tumaco) [13]. LD, STRUCTURE and $F_{ST}$ analyses all rely on allelic variation. The concordance between results based on relatedness and allelic variation suggests that *P. falciparum* outbreeding on the Colombian-Pacific coast is infrequent enough that both types of analyses generate insight on approximately the same time scale.

The aforementioned results generate hypotheses around the frequency of marine traffic and malaria transmission on the Colombian-Pacific coast. Notwithstanding long-range wind-borne dispersal, which may be critical for malaria transmission in Africa [49], anopheline flight range is generally small (around 3.5 km [50]). As such, long-range malaria parasite dispersal on the Colombian-Pacific coast is almost certainly human-mediated. A recent study of *P. vivax* proposed that human movement across a "malaria corridor" stretching from the northwest to the south of the Colombian-Pacific Coast likely promotes *P. vivax* gene flow, and that mining activities may provide transmission "contact zones" [51], similarly proposed for *P. falciparum* [22]. *P. falciparum* population connectivity is consistent with the human "malaria corridor" hypothesis, especially since it correlates with accessibility, not isolation-by-distance. Both infected humans and mosquitoes are compatible with this hypothesis, i.e. checks for infected *Anopheles spp.* on boats may be merited [52, 53]. However, relatively high differentiation between populations of *An. albimanus* (one of the three primary vectors of malaria in Colombia [54]) from Buenaventura and Tumaco [55] points towards human carriage.

The Colombian-Pacific coast has long been associated with international trade, but until recently human migration in the region was largely domestic. The flow of Venezuelan migrants infected with *Plasmodium spp.* has increased in recent years: of 965, 1774 and 2288 non-domestic malaria cases reported in Colombia in 2017, 2018 and 2019, respectively, 882 (91.4%), 1684 (94.9%), and 2190 (95.7%) were from Venezuela [56–58]. Other non-domestic sources of malaria in Colombia include countries elsewhere in South America (e.g. Peru, Panama, French Guyana, Ecuador, Brazil) and several African countries (e.g. Uganda, Republic of the Congo, Nigeria, Ivory Coast, Cameroon, Angola) [58]. Some of the infected Venezuelan nationals are migrating southward to Ecuador and Peru [24]. Other non-domestic cases may be associated with the traffic of people who arrive at Colombian ports with a view towards northward travel e.g. to the USA via Central America and Panama [59]. Genetic surveillance of "international parasites" may help malaria control efforts in Colombia.

The evidence we find of connectivity between *P. falciparum* populations may be unique to the period of time over which the data were collected (1993-2007). This was a period of historically high malaria case counts in Colombia [17], as well as social instability in the South Pacific region. Contemporary data on more densely sampled cases and on mosquito and human movement are required to characterise extant connectivity, its reach beyond Colombia (see e.g. [47]), and to rule out alternative hypotheses. Regarding alternative hypotheses,

heterogeneous vectorial capacity and antimalarial drug pressure could selectively enhance parasite survival in such a way that generates apparent connectivity between Buenaventura and Tumaco, e.g. if parasites are adapted to local vectors whose distributions are more similar between Buenaventura and Tumaco than elsewhere. Although adult *An. albimanus* B and *An. neivai* s.l. have been detected in the vicinities of both cities [55, 60], the species distributions in the vicinities of Buenaventura and Tumaco differ more than those in the vicinities of Tumaco and Guapi [60]. As such, heterogeneous vectorial capacity seems an unlikely alternative hypothesis. Similarly, relatedness may be greater among parasites with comparable antimalarial resistance: a recent study of South East Asian *P. falciparum* parasites found greater relatedness in the recent past among parasites with artemisinin resistance mutations versus without [61]. This study used size-stratified IBD segments to date relatedness [61]. On the Colombian-Pacific coast, IBD segment size inference could help identify some recently related parasites. However, it requires whole genome sequence data and is hard (if not presently impossible) to interpret in the face of frequent clonal propagation [32]. The development of an ancestral recombination model that incorporates transmission-dependent selfing is a research priority in malaria genomic epidemiology and would aid research on other organisms that show both outbreeding and clonal propagation.

## Materials and methods

### Data

This study relies entirely on previously published data that are publicly available [13, 32]. In the original study by Echeverry et al., finger-prick blood spot samples were obtained from patients with symptomatic uncomplicated malaria [13]. Samples were collected between 1993 and 2007 from five cities in four provinces: Tadó and Quibdó in Chocó, Buenaventura in Valle, Guapi in Cauca and Tumaco in Nariño (S1 Table) [13]. Informed consent was obtained from all the subjects enrolled, as approved by the CIDEIM Institutional Review Board (IRB) [13]. The Colombian-pacific coast is one of the rainiest regions of the world [55, 62]. At that time, Colombia had approximately 100,000 malaria cases per year [13, 17]. Collectively Chocó, Valle, Cauca and Nariño accounted for up to 75% of the *P. falciparum* cases reported, with relatively high transmission in Chocó and relatively low transmission in Valle and Cauca [13].

The data that feature in this descriptive study also feature in a recent methodological study concerning data requirements for relatedness inference [32]. As in [32], we did not post-process the data in any way besides mapping SNP positions to the *P. falciparum* 3d7 v3 reference genome and recoding heteroallelic calls as missing (since all samples with fewer than 10 heteroallelic SNP calls were classified monoclonal previously [13]). The monoclonal data include 325 *P. falciparum* samples with data on 250 biallelic SNPs whose minor allele frequency estimates (the minor allele sample count divided by 325) range from 0.006 to 0.495 (S4 Fig).

### Relatedness inference and classification of parasite sample pairs and groups

For each pairwise parasite sample comparison, we generated a relatedness estimate and 95% confidence interval using the HMM and parametric bootstrap described in [32]. Sample pairs were classified as unrelated, related, highly-related and clonal using confidence interval endpoints as follows and summarised in Table 1. A pair was classified unrelated if its relatedness estimate, $\hat{r}$, was statistically indistinguishable from zero with lower confidence interval endpoint (LCI) less than $\epsilon$, an arbitrarily small number to identify LCI $\approx 0$ and UCI $\approx 1$ given that LCI and UCI $\in (0, 1)$ not $[0, 1]$. A pair was classified related if its relatedness estimate, $\hat{r}$, was

statistically distinguishable from zero with LCI $> \epsilon$. A pair was considered highly-related if its relatedness estimate, $\hat{r}$, was statistically distinguishable from some specified threshold, $\tau$, with LCI $> \tau$. A pair was considered clonal if its relatedness estimate, $\hat{r}$, was statistically indistinguishable from one with upper confidence interval end-point (UCI) $> 1 - \epsilon$. Note that these classifications are possible because all estimates are informative, i.e. no confidence intervals span the entire zero to one range (Fig 1). These classifications are neither necessarily exclusive nor conversely true: a clonal parasite pair is related, but a related parasite pair is not necessarily clonal. Throughout, $\epsilon = 0.01$. In the main analysis (Fig 2) $\tau = 0.25$, in the sensitivity analysis (S1 Fig) $\tau \in \{0.25, 0.50\}$.

In addition to classifying parasite sample pairs, we classify groups of statistically indistinguishable parasite samples, which we call clonal components because they are defined using the simple concept of components from graph theory. First, we construct a super-graph whose vertices are parasite samples connected by edges that are weighted by relatedness estimates. Within the super-graph, a clonal component is a sub-graph within which all parasite samples are connected to one another (directly or not) via edges whose weights are statistically indistinguishable from one, while being connected to parasites samples outside the sub-graph via edges whose weights are not statistically indistinguishable from one. Clonal components tend to be fully connected (i.e. all parasite samples within the clonal component are directly connected to one another by edges whose weights are statistically indistinguishable from one). The `igraph` package [63] in R [64] was used to identify clonal components and to visualise them using the Fruchterman-Reingold layout algorithm [45].

## Spatiotemporal trends in *P. falciparum* population connectivity

Spatiotemporal trends in population connectivity were explored visually by partitioning parasite sample pairs by their collection cities and dates, then plotting the per-partition fraction of highly-related pairs. Inter-city great-circle distance was calculated using the Haversine formula, which assumes the earth is spherical. Error bars were constructed by re-sampling per-partition parasite sample pairs 100 times with replacement and taking the 2.5th and 97.5th percentiles of the fraction of highly-related pairs as the lower and upper limits, respectively. Sensitivity to $\tau = 0.25$ (high relatedness threshold used in Fig 2) was explored using an alternative $\tau = 0.50$ (S1 Fig) and also by using a threshold-free approach (Fig 3) as follows.

To explore population connectivity using a threshold-free approach, we calculated 1-Wasserstein distances between groups of parasite samples from different cities using the `transport` [65] package in R [64]. Specifically, for a pair of cities $a$ and $b$, we construct a $n_a \times n_b$ genetic distance matrix, $G$, of $1 - \hat{r}_{ij}$ (where $n_a$ and $n_b$ are the parasite sample counts from cities $a$ and $b$, respectively, $i = 1, \ldots, n_a$ and $j = 1, \ldots, n_b$) and two vectors $w_a = \left(\frac{1}{n_a}, \ldots, \frac{1}{n_a}\right)$ and $w_b = \left(\frac{1}{n_b}, \ldots, \frac{1}{n_b}\right)$ of length $n_a$ and $n_b$, respectively. We then calculate the 1-Wasserstein distance, which minimises the total cost of transporting $w_a$ to $w_b$, where $1 - \hat{r}_{ij}$ is the cost of transporting a single unit, using `transport::transport(`$w_a$`, `$w_b$`, costm = `$G$`, method = "shortsimplex")`. This amounts to treating parasite samples from different cities as draws from different distributions, where the 1-Wasserstein distance can be interpreted as the cost required to transport a distribution of parasite samples from one city to another [18, 34]. Since per-city parasite sample sizes differ, transportation requires the expansion (or contraction) of parasite mass in addition to the transportation of individual units. City pairs with smaller 1-Wasserstein distances are interpreted as having greater connectivity between the *P. falciparum* populations collected from them. Error bars were constructed by re-sampling parasite sample pairs per inter-city partition 100 times with replacement and taking the 2.5th and

97.5th percentiles of the distribution of 1-Wasserstein distances based on the re-sampled sample pairs as the lower and upper limits, respectively.

## Supporting information

**S1 Table. Yearly monoclonal *P. falciparum* sample counts per city.**
(PDF)

**S2 Table. A summary of all clonal components.** Date and city refer to the date and city of collection of the earliest parasite sample per clonal component.
(PDF)

**S3 Table. Average relatedness between select clonal components.** Average relatedness to three decimal places between clonal components (CCs) 1, 12, 14, 20 and 40 with the maximum 2.5% end-point of the 95% confidence intervals per CC in parentheses. The maximum 2.5% end-point indicates that relatedness between C20 and C40 is not statistically distinguishable from zero, for example.
(PDF)

**S1 Fig. Fractions of highly-related sample pairs partitioned in time and space: Sensitivity to the high-relatedness threshold.** Highly-related samples pairs are defined as those with lower confidence interval end-point (LCI) of relatedness estimate, $\hat{r}$, greater than thresholds 0.25 and 0.50; or with upper confidence interval end-point (UCI) of $\hat{r} > 0.99$ (i.e. clonal parasite sample pairs). Colours correspond to Fig 1. (A) Partitioned by time between collection dates. (B) Partitioned by collection city.
(TIF)

**S2 Fig. Sample collection chronology does not reflect transmission chain event chronology.** Schematic illustrating why sample collection chronology is not necessarily representative of the true sequence of transmission chain events when sampling is sparse and clonal propagation is frequent. The schematic shows two hypothetical locations A and B where malaria parasites have been sampled sparsely: solid ellipses represent sampled parasites, open ellipses represent parasites that were present but not sampled, different colours denote different parasite genotypes.
(TIF)

**S3 Fig. Clonal components and singletons and the average relatedness between them.** Vertices depict clonal components (CCs), which are groups of two or more statistically indistinguishable parasite samples, and singletons, which are individual parasite samples that do not belong to a CC. Vertices are plotted using the Fruchterman-Reingold layout algorithm [45], thereby clustering inter-related vertices. The size of each CC vertex is proportional to the number of parasite samples per CC, ranging from 2 to 28 statistically indistinguishable parasite samples. CCs are named in order of the collection date of the earliest parasite sample per CC. CCs with parasite samples from two or more sites are depicted as pie charts. Colour denotes the city of parasite sample collection. Edge transparency and weight is proportional to average relatedness, ranging from 0.003 to 0.912. Relatedness estimates that are indistinguishable from zero were set to zero. Edges whose average relatedness is zero are not plotted. Each CC besides CC4 is related to at least one other. CC4 is likely a contaminant; see main text. A singleton from Buenaventura, which is loosely related to CC4, may also be a contaminant.
(TIF)

**S4 Fig. Minor allele frequency estimates.** Histogram of minor allele frequencies estimated using all 325 monoclonal *P. falciparum* samples genotyped at 250 biallelic SNPs.
(TIF)

## Acknowledgments

Thank you to Pierre Jacob for guidance on the calculation of the 1-Wasserstein distances and to James Watson, Manuela Carrasquilla and Vladimir Corredor for helpful comments and discussion.

## Author Contributions

**Conceptualization:** Aimee R. Taylor.

**Data curation:** Aimee R. Taylor, Diego F. Echeverry.

**Formal analysis:** Aimee R. Taylor.

**Funding acquisition:** Diego F. Echeverry, Timothy J. C. Anderson, Daniel E. Neafsey, Caroline O. Buckee.

**Investigation:** Diego F. Echeverry, Timothy J. C. Anderson.

**Methodology:** Aimee R. Taylor.

**Resources:** Diego F. Echeverry, Timothy J. C. Anderson.

**Supervision:** Daniel E. Neafsey, Caroline O. Buckee.

**Visualization:** Aimee R. Taylor.

**Writing – original draft:** Aimee R. Taylor.

**Writing – review & editing:** Aimee R. Taylor, Diego F. Echeverry, Timothy J. C. Anderson, Daniel E. Neafsey, Caroline O. Buckee.

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
