## [Decision Letter · Decision Letter 0]

1 Jun 2020

Dear Dr Taylor,

Thank you very much for submitting your Research Article entitled 'Identity-by-descent relatedness estimates with uncertainty characterise departure from isolation-by-distance between Plasmodium falciparum populations on the Colombian-Pacific coast' to PLOS Genetics. Your manuscript was fully evaluated at the editorial level and by independent peer reviewers. The reviewers appreciated the attention to an important topic but identified some aspects of the manuscript that should be improved.

We therefore ask you to modify the manuscript according to the review recommendations before we can consider your manuscript for acceptance. Your revisions should address the specific points made by each reviewer.

[LINK]

Yours sincerely,

Giorgio Sirugo

Associate Editor

PLOS Genetics

Gregory Barsh

Editor-in-Chief

PLOS Genetics

Reviewer's Responses to Questions

**Comments to the Authors:**

Reviewer #1: The authors perform identity-by-descent (IBD) analysis on genotype data for 325 P. falciparum samples in order to understand Malaria transmission in five cities along the Colombian-Pacific coast. Results from this analysis compliment and expand on a previous study of this cohort which used alternative techniques to examine transmission along the coast. The authors provide evidence of greater diversity in P. falciparum genomes in cities with heavy marine traffic, as well as greater gene flow between cities with marine ports than those without.

Results from this analysis suggest a need for greater precautions at wharfs and ports along the Colombian-Pacific coast to reduce Malaria transmission via marine traffic. This is a well written manuscript that uses statistically sound and logical analyses, where the results are supported by geographical and marine traffic data. I also commend the authors on their original and informative graphics. I have only minor comments.

Minor comments

• In figures 5 and A.2, the samples from Tado and Quibdo are highly related within and between these two cities, even though there is ~3 years difference the collection date between the samples from Tado and Quibdo (e.g. CC10 and CC34). I assume the geographical location/isolation of Tado and Quibdo may explain the genotypic similarity. Would this also explain why the samples have remained so genetically similar over time, or could there be another explanation for this?

• Line 55 page 2: I cannot make sense of the following sentence, please re-write. “As such, analyses of relatedness can sometimes nearby and recent connectivity where analyses of FST cannot.”

Reviewer #2: This is a well-written ms detailing the development and application of IBD relatedness approaches in malaria populations sampled along the Pacific Coast of Colombia. As the authors state relatedness approaches including those which utilise IBD approaches are widely used but rarely, to my knowledge, provide estimates of uncertainty. I would think that the methods described in the ms should be of relevance to a broad swathe of thr PLoS Genetics readership, particularly those interested in conservation biology/ genomic epidemiology in recombining organisms. Most of my comments are of a picayune/ muggen ziften nature with possible exception of comments starting "Line 149" and "Line 163."

Line 41 distinct rather than disparate (a difference in essence)

Line 45 clarify which “them”

Line 55 something missing here

Line 60 “In any event” superfluous

Line 63 Although transmission of distinct parasite clones could be the result of separate infection events over the course of the mosquito life span

Line 74 sentence “Departure….” Partial repeat of previous

Line 77 solely in malaria epid or more generally

Line 79 curious phrasing “our response to it”

Line 83 is it critical? Does drug policy change at a sub-national level? -Needs more support for such a broad statement

Line 88 direction of clonality and incidence relationship- presume inverse?

Line 102 require a little more explanation of what data the threshold was based upon eg % SNPs shared in pairwise comparisons

Line 106 “optimal transport using Wasserstein distance” will read as jargon to the casual reader (and to me)

Line 111 not clear what brittle means in this sentence

Line 115 something missing here “plan to”?

Line 118 State which relatedness estimator was used

Figure 1 more detail in figure legend required eg what’s tau? IN general I find the figure legends a little brief. Always feel that figures should be comprehensible without reference to the text

Table 1 similarly define r hat; epsilon etc

Line 124 capital H?

Line 125 define why is this a high relatedness threshold?

Figure 2 colour contrast in panel b is poor. Also font size of legend and axis labels too small. Is distance measured by simple straight line or by some other means eg via transport infrastructure

Line 128 replace far apart with space

Line 129 replace versus with than

Line 131 exceptionally high? Perhaps greater than expected given geographic distance

Line 135 meaning of “effort required” is unclear

Line 142 capitalise pan-american

Line 149 Can you test this supposition by analysing data within years or at least within the 2004-2206 collection window from which most samples derive to demonstrate elevated relatedness in temporally proximate collections?

Line 160 I think a small expansion is required here rather then referring to methodology. This sentence cannot be readily parsed by the general reader

Line 163 how is clonal relatedness defined

Line 163 How can you discount a rare importation event from genetically distinct population? Obviously this putative contamination event was only detected as the clonal component was so different. I would imagine that contamination events at time of collection would inflate estimates of relatedness within cities and within time points. How would you refute this suggestion?

Fig 3 same issue with font size

Figure 4b perhaps overlay city labels? The zoom feature did not work on my paper print out.

Line 173 would be good to see some data discussing MOI in these populations to support this assertion

Figure 6 Not sure this is necessary I doubt anyone will dispute that without sampling to exhaustion not all haplotype will be detected in a single collection period

Line 196 verbatim repeat of sentence in introduction

Line 226 “gold and narcotics” is this necessary? Could be seen to be perpetuating a stereotype. Would international trade suffice?

Line 229 Second sentence does not support the assertion that flow of infected migrants has increased, just reveals that they come predominantly form Venezuela

Line 236 Not trying to sanitise the ms but again “considerable violence” could this be rephrased- social instability—as written implies considerable violence is an innate property of the south pacific region rather than reflective of societal breakdown; malign foreign influence etc. Also why would high malaria case counts result in fleeting connectivity?

References are inconsistently formatted

Reviewer #3: This manuscript presents a nicely done reanalyses of a data set based on 13-27 year old samples from P. falciparum infection from the Pacific Coast of Colombia. Using the combination of IBD-based analytical approaches the authors demonstrated the importance of martime traffic in shaping the population structure.

While they may have not been applied in such a way to malaria epidemiology, the methods in themselves are not entirely novel. Similarly, why they very nicely and convincingly demonstrate the very particular nature malaria transmission in the study area, this does not lead to a fundamental change in our understanding local malaria epidemiology, rather it re-inforces patterns that have been previously recognised.

The authors claim broad relevance of their IBD-based approaches by pointing out that they could be applied other organisms with mixed mating systems. It would have been much more interesting if they had actually demonstrated that broad relevance by applying their methods either further, more recent malaria dataset (with different underlying transmission patterns) or to other organisms. Without such a demonstration, the manuscript remains just a nicely done case study of the power IBD-based analytical methods.

**Have all data underlying the figures and results presented in the manuscript been provided?**

Reviewer #1: Yes

Reviewer #2: Yes

Reviewer #3: Yes

PLOS authors have the option to publish the peer review history of their article (what does this mean?). If published, this will include your full peer review and any attached files.

Reviewer #1: No

Reviewer #2: No

Reviewer #3: No

---

## [Editor Report · Decision Letter 1]

8 Sep 2020

Dear Dr Taylor,

We are pleased to inform you that your manuscript entitled "Identity-by-descent relatedness estimates with uncertainty characterise departure from isolation-by-distance between Plasmodium falciparum populations on the Colombian-Pacific coast" has been editorially accepted for publication in PLOS Genetics. Congratulations!

Yours sincerely,

Giorgio Sirugo

Associate Editor

PLOS Genetics

Gregory Barsh

Editor-in-Chief

PLOS Genetics

Comments from the reviewers (if applicable):

**Data Deposition**

http://datadryad.org/submit?journalID=pgenetics&manu=PGENETICS-D-20-00480R1

**Press Queries**

---

## [Editor Report · Acceptance letter]

6 Nov 2020

PGENETICS-D-20-00480R1 

Identity-by-descent with uncertainty characterises connectivity of *Plasmodium falciparum* populations on the Colombian-Pacific coast 

Dear Dr Taylor, 

We are pleased to inform you that your manuscript entitled "Identity-by-descent with uncertainty characterises connectivity of *Plasmodium falciparum* populations on the Colombian-Pacific coast" has been formally accepted for publication in PLOS Genetics! Your manuscript is now with our production department and you will be notified of the publication date in due course.

With kind regards,

Matt Lyles

PLOS Genetics

On behalf of:
